# How electronic health literacy influences physical activity behaviour among university students: A moderated mediation model

Ning Zhou[1‡], Fanzheng Mu[1‡], Yao Zhang[1], Dingyou Zang[1], Weidong Zhu[1], Xiaoyu Wang[1], Wei Wang[1], Haoyu Li[1], Jiaqiang Wang[1], Xingyu Zhang[1], Chenxi Li[1], Yuhan Li[1], Mohan He[1], Wenhao Zhang[1], Qi Liu[1], Bochun Lu[1], Shanshan Han[1], Yaxing Li[2], Yangsheng Zhang[3], Lingli Xu[1], Yuyan Qian[1], Lei Ding[1], Chuanyi Xu[4], Han Li[5], Shuo Feng[6], Lanlan Yang[1], Yong Wei[7,8], Bo Li[1]*

1 Institute of Sports Science, Nantong University, Nantong, China, 2 Physical Education University, Shangqiu University, Shangqiu, China, 3 School of Physical Education, Nanjing Xiaozhuang University, Nanjing, China, 4 Guangxi University of Chinese Medicine, Nanning, China, 5 Department of Physical Education, Ordos Institute of Applied Technology, Ordos, China, 6 University of Physical Education, Xinyang Normal University, Xinyang, Henan, China, 7 Nantong Institute of Technology, Nantong, Jiangsu, China, 8 Nantong Rehabilitation Hospital, Nantong, Jiangsu, China

‡ These authors contributed equally to this work.
* wangqiulibo@163.com

# Objective

## Purpose

This study examines how eHealth literacy influences exercise behaviour among university students through a moderated mediation model. Specifically, peer relationships are positioned as mediators, while sleep quality moderates the effect of eHealth literacy on exercise behaviour among university students. The study explores the intricate interactions involved and elucidates how eHealth literacy affects exercise behaviour through multiple dimensions.

## Methods

Student data were collected via questionnaire surveys across multiple academies. A stratified purpose sampling approach was employed for participant selection, targeting four specific universities to ensure regional and typhological representation: Nanjing Xiaozhuang University, Yangzhou University, Shangqiu University, and Yangzhou Polytechnic Institute. This process resulted in 14,892 valid responses for analysis, and SPSS 26.0 along with the PROCESS macro was used to analyses variables such as electronic health literacy, physical activity, sleep quality, and peer relationships.

**Data availability statement:** All relevant data are within the paper and its Supporting Information files.

**Funding:** The funder (2024 General Project of Philosophy and Social Science Research at Universities in Jiangsu Province, NO: 2024SJYB1253) only provide financial support for research and distribution of questionnaires, and are not involved in any research activities.

**Competing interests:** The authors have declared that no competing interests exist.

**Abbreviations:** Ehealth: Electronic health; SDT: Self-Determination Theory; CPAHLS-CS: Chinese University Students Physical Activity and Health Longitudinal Survey; eHLS: EHealth Literacy Scale; PARS-3: Physical Activity Rating Scale; PSQI: Pittsburgh Sleep Quality Index.

## Results

The structural equation modeling revealed a significant direct effect of eHealth literacy on university students physical activity behaviour *(β=0.065, p<0.001)*. Mediation analysis delineated an indirect pathway through peer relationships: eHealth literacy significantly enhanced peer relationship quality *(β=0.251, p<0.001)*, which subsequently predicted increased physical activity *(β=0.058, p<0.001)*. Peer relationships significantly mediate between eHealth literacy and university students physical activity *(β=0.251, p<0.001)*. Additionally, sleep quality serves as a moderating variable *(β=0.029, p<0.001)*, significantly moderating the direct effect of eHealth literacy on university students physical activity, with the 95% confidence intervals consistently excluding zero.

## Conclusion

EHealth literacy among university students can significantly and positively predict their exercise behaviour. Peer relationships mediate the effect of eHealth literacy on exercise behaviour. At the same time, sleep quality is a moderating variable influencing eHealth literacy's direct impact on university students exercise behaviour.

## Introduction

In the wake of the digital wave, eHealth literacy has emerged as a critical dimension in modern health promotion research [1]. It is defined as the comprehensive ability of individuals to obtain, understand, evaluate, and apply health information through electronic means. As of 2024, the internet penetration rate in China has surpassed 78.0%, and the primary channel for the public to access health information has shifted to digital platforms [2]. In the context of this technological transformation, university students, as digital natives, exhibit a significant disparity in their levels of eHealth literacy. Nationwide screening data reveal that merely 23.0% of this demographic meet proficiency thresholds, starkly contrasting the health literacy demands of an increasingly digitized society. Notably, while exercise behaviour is a key indicator of health promotion, the interactive mechanisms between exercise behaviour and eHealth literacy have yet to be thoroughly elucidated [3,4]. This theoretical ambiguity substantially impedes the development of targeted health intervention strategies.

From the Self-Determination Theory (SDT) perspective, the development of health behaviour necessitates fulfilling three psychological needs: autonomy, competence, and relatedness [5,6]. This theoretical framework provides critical insights for deciphering the interplay between eHealth literacy and physical activity. Existing research has demonstrated that participation in physical activity can enhance individuals intrinsic motivation for health management [7], facilitating the acquisition and application of eHealth information [8]. However, current studies concentrate on the direct correlation between eHealth literacy and exercise behaviour [9,10], lacking a systematic analysis of the mechanisms that link the two. Notably, there remains a significant theoretical gap regarding the mediating and moderating effects

of social interaction factors (such as peer relationships) and environmental factors (such as sleep quality) [11,12]. Sleep was chosen as a moderating variable because it forms the physiological basis for health information processing, while peer relationships bridge the knowledge-behaviour transformation through social support mechanisms. Together, these factors help to explain the boundary conditions and pathways through which electronic health literacy affects behaviour. This uni dimensional research paradigm is inadequate for elucidating the complex decision-making processes underlying behaviour, thereby limiting the effectiveness of health intervention strategies.

The moderating effect of sleep quality operates through dual physiological and psychological mechanisms: Cognitive neuroscience research has confirmed that adequate sleep enhances the activity level of the prefrontal cortex, thereby improving information processing efficiency [13,14]. Conversely, sleep deprivation weakens hippocampal function, leading to a decline in the capacity to evaluate health information [15]. Crucially, this neural-level processing advantage exhibits synergistic relationships with socio-cognitive operations. Simultaneously, the mediating role of peer relationships aligns with the framework of social cognitive theory—through observational learning and vicarious reinforcement mechanisms [16], group interactions can effectively enhance the ability to discern health information and convert social support into operational efficacy in eHealth literacy [17–19]. These findings underscore the necessity of constructing cross-level dynamic interaction models that explicitly address the coupling relationships between physiological and social systems.

Previous research exhibits three methodological constraints: first, cross-sectional designs fail to capture temporal dynamics between variables; second, self-reported measures are vulnerable to social desirability bias [20]; third, conventional mediation-moderation analyses predominantly adopt compartmentalized approaches, lacking multilevel systems modeling frameworks. This study innovative ly integrates the mediating pathway of peer relationships with the moderating effect of sleep quality into a unified model, utilizing structural equation modelling to validate the transmission mechanism of "literacy—social support—behaviour" and the boundary conditions of "literacy—environmental moderation—behaviour." This approach transcends the analytical dimensions of traditional research [21,22].

This study constructs a moderated mediation model, proposing a three-tiered hypothesis system: (1) eHealth literacy positively predicts exercise behaviour (H1); (2) peer relationships mediate the relationship between literacy and behaviour (H2); and (3) sleep quality moderates the first half of the mediation path (H3). The model will be tested using hierarchical regression and the Bootstrap method. The research findings aim to deepen our understanding of the mechanisms underlying the formation of health behaviour in the digital age, providing a dual pathway for health promotion programmed targeting university students—enhancing individual agency by improving eHealth literacy, while also optimizing sleep management and peer support to strengthen the environmental support system. The hypothesized model is illustrated in S1 Fig.

## 1 Methods

### 1.1 Data sources

This study employs data from the "Chinese University Students Physical Activity and Health Longitudinal Survey" (CPAHLS-CS). The recruitment period for the study spanned from 10th September 2023–30th September 2023, with the subsequent questionnaire survey being conducted from 10th September 2023–22th October 2023. The CPAHLS-CS is designed to collect a high-quality set of individual micro data representing Chinese university students physical activity and health behaviour. Its purpose is to analyses the intersection of physical activity and health among this demographic and to promote interdisciplinary research on health issues affecting university students. The CPAHLS-CS has been widely used to assess the health status of Chinese university students. The participants selected for this study are current students enrolled in conventional higher education institutions in mainland China, with the list of institutions referenced from the Ministry of Education's "List of National Conventional Higher Education Institutions (as of June 15, 2023)."Informed consent was obtained from all the subjects involved in the study. The study questionnaire followed the American Psychological Association (APA) Ethical Principles of Psychologists and Code of Conduct (APA Ethical Principles of Psychologists

and Code of Conduct). The study was completed anonymously. The collection of general demographic information for this study included school, gender, and grade level. Before the researcher administered the questionnaire, the researcher read the beginning of the questionnaire and informed the participant of the relevant informed consent and completion requirements, and after the participant verbally agreed, the researcher administered the questionnaire. After the participant read the beginning of the questionnaire again, if the participant was willing to continue to fill in the questionnaire, he could continue to fill in the questionnaire on a completely voluntary basis and his data would be collected. If the participant does not wish to complete this questionnaire, they can choose not to continue to complete the questionnaire, and we will not record their information. Before the study commenced, all participants were provided with a detailed explanation of the study's purpose, specific procedures, anticipated duration of participation, and any potential risks and discomforts by trained researchers. The researchers ensured that participants fully understood the relevant information and had the opportunity to ask questions to address any concerns. Upon confirming that participants comprehended the information and voluntarily agreed to take part, they expressed their consent orally. Furthermore, this study received approval from the Institutional Review Board (IRB) of Nantong University, with approval number 70.

## 1.2 Sampling methods

Survey participants were selected using stratified, cluster, and phased sampling methods.

**1.2.1 Determining sampling locations.** To ensure the representatives of the monitoring subjects, an average of three sampling locations was allocated across each province (region/city). Based on the principle of extracting equal samples from different cities, the specific approach was as follows: sampling locations were determined from the prefecture-level cities administered by each province or autonomous region. Capital cities were classified as "Category One" sampling locations. The other two sampling locations were identified based on the geographical positioning of the province or autonomous region, with one city representing a region of average socio-economic development classified as "Category Two" and a city with relatively poor social and economic development classified as "Category Three." In the case of municipalities directly under the central government, the sample selection primarily relied on random cluster sampling but also adhered to the principle of maintaining the quantity of the three sampling locations.

**1.2.2 Determining sampling units.** When selecting sampling units, three main considerations were taken into account: firstly, the higher education institutions selected must be officially registered universities as per the Ministry of Education, including vocational and technical colleges; secondly, the units must meet the sampling criteria (such as age, number of students, and grade distribution); thirdly, there should be a designated person responsible for distributing the questionnaire at the institution, which is willing to participate in the monitoring over the long term.

**1.2.3 Grouping and sample size.** The population was divided into two groups based on gender (male and female) and further categorized into eight sample types according to grade, with a minimum sampling size of 45 individuals per sample type (for example: first-year male students). The total sample size for each province (region or municipality) was set at 1,080 people, with an estimated total of 33,480 participants completing the questionnaire nationwide (excluding Hong Kong, Macau, and Taiwan). In September 2023, an electronic survey was conducted using the Questionnaire Star software across administrative classes, resulting in the collection of 36,756 completed questionnaires.

**1.2.3 Second-stage sampling.** This secondary sampling collected data from students at four institutions: Nanjing Xiao Zhuang University, Yangzhou University, Shangqiu University, and Yangzhou Industrial Vocational Technical University. Selection criteria for the institutions: 1) Regional representation (East China/ Central China); 2) Coverage of types of education (comprehensive universities/ normal universities/ application-oriented undergraduate institutions/ vocational colleges); 3) Exclusion of sports-related institutions to avoid sample bias. The minimum sample size calculation was conducted using the WJX (Questionnaire Star) software, yielding 15,805 valid questionnaires. The minimum sample size was determined using S2 Fig, with the Type I error rate set at 0.05 and the allowable margin of error set at 0.01, while the sample proportion (P) was established at 0.05. Following inquiries on the official websites of the four institutions, the

total number of enrolled students was 110,700 (data updated for 2024). Consequently, the finite population size (N) was set at 110,700, resulting in a calculated minimum sample size of 1,107. In this study, the data were filtered according to the following criteria: ① Delete questionnaires completed by respondents aged under 18 or over 23; ② Exclude incomplete questionnaires or those with missing responses; ③ Remove entries with contradictory or conflicting options; ④ Discard multiple submissions from the same IP/ID; ⑤ excluding questionnaires that did not follow the instructions and had consistent answers to all questions, resulting in a total of 913 questionnaires being removed. After sample screening, a total of 14,892 valid questionnaires were retained, resulting in a valid response rate of 95%. The overall sample meets the minimum sample size requirements. The distribution of the sample is presented in Table 1.

## 1.3 Variable measurement

### 1.3.1 EHealth literacy scale (eHLS).

Ehealth literacy among university students was measured using the eHealth Literacy Scale (eHLS) developed by Norman and Skinner [23]. This scale encompasses three dimensions: the ability to access, evaluate, and apply health information and services found online, comprising a total of eight items. The scoring method employed a 5-point Likert scale, where responses ranged from "Strongly Disagree" (1 point) to "Strongly Agree" (5 points). Higher scores indicate a greater level of eHealth literacy. Previous studies have demonstrated that the eHLS possesses high reliability and validity among Chinese university students, with a Cronbach's α coefficient of 0.870. The Cronbach's α coefficients for the three dimensions ranged from 0.810 to 0.860 [24,25].

### 1.3.2 Physical activity rating scale (PARS-3).

This study utilities the Physical Activity Rating Scale (PARS-3), initially developed by Japanese scholar Hashimoto and revised by Liang et al., to assess the physical activity levels of university students. The PARS-3 evaluates three aspects of physical activity: intensity, frequency, and duration of each exercise session, thus measuring students overall participation in physical activities. In the questionnaire, each item is rated on a 5-point scale, ranging from "1" (never engages in physical exercise) to "5" (regularly participates in physical exercise). Higher scores indicate greater levels of physical activity, with the results serving as a metric for the quantity of exercise behaviour, thereby reflecting university students participation in physical activities within a specific time frame. The raw scores obtained from the questionnaire are calculated using the formula: ***physical activity score = intensity x (duration per session) x frequency [26]***. The normative classifications for physical activity among Chinese adults are low activity ≤19 points, moderate activity 20–42 points, and high activity ≥43 points. The test-retest reliability of this scale is reported to be 0.820 [27].

### 1.3.3 Pittsburgh sleep quality index (PSQI).

The Pittsburgh Sleep Quality Index (PSQI), developed by Buysse et al. in 1989, measured sleep quality among university students [28]. This scale consists of 19 self-rated and five observer-rated items, with the 19th and five observer-rated items excluded from scoring. The 18 self-rated items contribute to seven factors: sleep quality, sleep onset time, sleep duration, sleep efficiency, sleep disturbances, use of sleeping medication, and daytime dysfunction. Each factor is scored from 0 to 3, resulting in a total score ranging from 0 to 21. A total score

**Table 1. Sample distribution table.**

| Variable | Option | Frequency | Percentage (%) |
|---|---|---|---|
| Gender | | | |
| | Male | 6346 | 42.6 |
| | Female | 8546 | 57.4 |
| Grade | | | |
| | Junior | 3624 | 24.3 |
| | Senior | 1087 | 7.3 |
| Total | 14892 | | 100 |

of ≤7 indicates good sleep quality, whereas a score of >7 signifies poor sleep quality, with higher scores indicating worse sleep quality. Previous studies have demonstrated that the PSQI possesses high reliability and validity among Chinese university students, with an overall Cronbach's α of 0.870 and a split-half reliability coefficient of 0.870 [29].

**1.3.4 Asher's peer relationship scale.** The Asher's Peer Relationship Scale, developed by Asher, was employed to measure peer relationships among university students, specifically assessing the quality of peer relationships in children and adolescents [30]. The scale comprises 16 items and utilities a four-point Likert scoring system ranging from 1 ("completely agrees") to 4 ("completely disagrees"). Higher scores indicate better peer relationships. Previous research has demonstrated that the eHealth literacy scale (eHLS) exhibits high reliability and validity among Chinese university students, with internal consistency reliability reflected by Cronbach's α of 0.870 [31].

## 1.4 Statistical methods

Data analysis was conducted using SPSS 26.0 and the PROCESS macro programmer. The analysis was carried out in several steps: 1) Common method bias testing was performed using Harman's single-factor test through factor analysis in SPSS; 2) Descriptive statistical analysis was conducted, utilizing independent samples t-tests to describe the demographic variables of the sample and the research variables; 3) Correlation analysis was performed using Pearson correlation coefficients to examine the relationships between variables; 4) Regression analysis was employed to investigate the influence of various variables on each other, to understand the relationships between variables, predict the values of the dependent variable, and assess the impact of the independent variable, eHealth literacy, on the dependent variable, exercise behaviour; 5) Mediation effect testing was conducted to examine the role of the mediating variable, peer relationships, in the relationship between eHealth literacy and exercise behaviour, while also assessing the moderating effect of sleep quality.

Given that the data on university students were collected through a questionnaire, which may be subjective, common method bias was assessed using Harman's single-factor test, control variable methods, and differential correction methods for exploratory analysis of the variables related to eHealth literacy, exercise behaviour, sleep quality, and peer relationships [32]. The results indicated that six principal components were extracted, with eigenvalues greater than 1, and the most significant factor explained 22.3% of the variance, surpassing the minimum criterion of 40.0%. Therefore, this study does not present notable standard method bias.

## 2 Results

### 2.1 Descriptive analysis

According to the results presented in Table 2, most university students engage in low levels of physical exercise, accounting for 67.2%. Regarding gender, the volume of physical exercise among females is significantly lower than that of males *(V=0.289, P<0.001)*, with a statistically significant difference. Low exercise levels average 80.0% at the grade level, whereas high exercise levels account for only 7.57%. A statistical analysis of the differences in physical exercise volume by gender across grades shows significant variations *(V=0.038, P<0.001)*. The distribution results indicate that third-year students have the highest proportion of low exercise levels, at 81.6%, while first-year students primarily exhibit a medium exercise level, accounting for 13.3%. Second-year students display the highest proportion of high exercise levels, at 8.99%. A descriptive statistical analysis of the survey data reveals that there are statistically significant differences in peer relationships by gender *(η²=0.001, P<0.001)*, with mean and standard deviation values of 41.580±7.993 for males and 41.980±6.353 for females. Aside from eHealth literacy *(P=0.002)*, the remaining three variables exhibit significant differences by gender *(P<0.001)*, with the most considerable disparity found in exercise behaviour between male and female university students *(η²=0.095)*. At the grade level, all variables, except for peer relationships *(P=0.006)*, show significant differences *(P<0.001)*, with the enormous difference across grades observed in eHealth literacy *(η²=0.006)*.

| Variable | Rating Level | Male(n=6346) n | % | Female (n=8546) n | % | Freshmen (n=4595) n | % | Sopho-more(n=5586) n | % | Junior(n=3624) n | % | Senior(n=1087) n | % |
|---|---|---|---|---|---|---|---|---|---|---|---|---|---|
| Physical Activity Level | | | | | | | | | | | | | |
| | Large Physical Activity | 932 | 14.7 | 197 | 2.31 | 275 | 5.99 | 502 | 8.99 | 265 | 7.31 | 87 | 8 |
| | Medium Physical Activity | 1152 | 18.2 | 714 | 8.36 | 613 | 13.3 | 717 | 12.8 | 401 | 11.1 | 135 | 12.4 |
| | Small Physical Activity | 4262 | 67.2 | 7635 | 89.3 | 3707 | 80.7 | 4367 | 78.2 | 2958 | 81.6 | 865 | 79.6 |
| | $x^2$ | | 1239.659 | | | | | 43.427 | | | | | |
| | p | | <0.001 | | | | | <0.001 | | | | | |
| | Gramer's V | | 0.289 | | | | | 0.038 | | | | | |
| Sleep Quality | | | | | | | | | | | | | |
| | Low Sleep Quality | 905 | 14.3 | 1268 | 14.8 | 565 | 12.3 | 858 | 15.4 | 584 | 16.1 | 166 | 15.3 |
| | High Sleep Quality | 5441 | 85.7 | 7278 | 85.2 | 4030 | 87.7 | 4728 | 84.6 | 3040 | 83.9 | 921 | 84.7 |
| | $x^2$ | | 49.244 | | | | | 53.504 | | | | | |
| | p | | <0.001 | | | | | <0.001 | | | | | |
| | Gramer's V | | 0.058 | | | | | 0.035 | | | | | |
| Ehealth Literacy | | 28.2 | 8.601 | 28.61 | 7.573 | 29.28 | 7.539 | 28.3 | 8.071 | 27.76 | 8.322 | 27.82 | 8.518 |
| $\eta^2$ | | | 0.001 | | | | | 0.006 | | | | | |
| F | | | 9.368 | | | | | 28.191 | | | | | |
| P | | | 0.002 | | | | | <0.001 | | | | | |
| Peer Relationship | | 41.58 | 7.993 | 41.98 | 6.353 | 41.96 | 5.985 | 41.55 | 7.557 | 41.99 | 7.439 | 41.91 | 7.808 |
| $\eta^2$ | | | 0.001 | | | | | 0.001 | | | | | |
| F | | | 11.782 | | | | | 4.123 | | | | | |
| P | | | <0.001 | | | | | 0.006 | | | | | |

## 2.2 Related analysis

The correlations among the variables are presented in Table 3. Exercise behaviour and sub-dimensions correlate significantly positively with eHealth literacy, with correlation coefficients ranging from 0.050 to 0.073. Conversely, exercise behaviour and its sub-dimensions are negatively correlated with sleep quality, with coefficients ranging from −0.075 to −0.053, suggesting that sleep quality indirectly impacts the enhancement of exercise levels. Furthermore, exercise behaviour and its sub-dimensions show a significant positive correlation with peer relationships, with coefficients ranging from 0.032 to 0.047. EHealth literacy is also significantly positively correlated with peer relationships *(r=0.251)*, indicating that both eHealth literacy and peer relationships significantly influence exercise behaviour. Additionally, a notable positive

**Table 3. Correlation analysis.**

| | Exercise Behaviour | Exercise Intensity | Exercise Duration | Exercise Frequency | EHealth Literacy | Sleep Quality | Peer Rela-tionships |
|---|---|---|---|---|---|---|---|
| Exercise Intensity | 0.783** | 1 | | | | | |
| Exercise Duration | 0.750** | 0.518** | 1 | | | | |
| Exercise Frequency | 0.461** | 0.230** | 0.391** | 1 | | | |
| EHealth Literacy | 0.065** | 0.073** | 0.060** | 0.050** | 1 | | |
| Sleep Quality | −0.056** | −0.053** | −0.075** | −0.070** | 0.012 | 1 | |
| Peer Relationship | 0.041** | 0.032** | 0.047** | 0.039** | 0.251** | −0.033** | 1 |

** Correlations are significant at the 0.01 level (two-tailed).

correlation between sleep quality and peer relationships *(r=0.012)* highlights the significant relationship between sleep and peer dynamics.

## 2.3 Regression analysis

The following model was established based on hierarchical regression analysis: physical exercise, peer relationships, and eHealth literacy were set as independent variables; gender and academic year were designated as control variables; and exercise behaviour was the dependent variable. The results of the chain mediation effect analysis are presented in Table 4. After controlling for gender and academic year, physical exercise positively predicted eHealth literacy *(β=0.073, SE=0.051, t=1.447)* and peer relationships *(β=0.077, SE=0.020, t=3.895, P<0.01)*, while negatively predicting sleep quality *(β=−0.201, SE=0.070, t=−2.860, P<0.05)*. Peer relationships positively predicted eHealth literacy *(β=0.223, SE=0.007, t=31.671, P<0.001)*. EHealth literacy positively predicted exercise behaviour *(β=0.021, SE=0.019, t=1.131)*, sleep quality *(β=0.029, SE=0.009, t=3.331, P<0.001)*, and peer relationships *(β=0.277, SE=0.011, t=25.238, P<0.001)*. Sleep quality negatively predicted exercise behaviour *(β=−0.020, SE=0.004, t=−5.523, P<0.001)*.

## 2.4 Mediation effect testing

According to the step wise regression analysis method for mediation effects proposed by Wen et al. (2004), the following models were established: In the first step, a regression analysis was conducted with exercise behaviour as the dependent variable and eHealth literacy as the independent variable, establishing Model 1 to examine the total effect of eHealth literacy on exercise behaviour (path c). In the second step, a regression analysis was performed with peer relationships

**Table 4. Regression analysis excluding confounding factors.**

| Regression Equation | | Overall Fit Index | | | Regression Coefficient Significance | | |
|---|---|---|---|---|---|---|---|
| Dependent Variable | Predictor Variable | $R$ | $R^2$ | $F$ | $β$ | $SE$ | $t$ |
| Exercise Behaviour | | 0.321 | 0.103 | 284.838*** | | | |
| | EHealth Literacy | | | | 0.073 | 0.051 | 1.447 |
| | Peer Relationship | | | | 0.077 | 0.020 | 3.895** |
| | Sleep Quality | | | | −0.201 | 0.070 | −2.860* |
| | Gender | | | | −10.866 | 0.275 | −39.556*** |
| | Grade | | | | 0.005 | 0.149 | 1.282 |
| Peer Relationship | | 0.253 | 0.064 | 338.582*** | | | |
| | EHealth Literacy | | | | 0.223 | 0.007 | 31.671*** |
| | Gender | | | | 0.320 | 0.114 | 2.806** |
| | Grade | | | | 0.155 | 0.062 | 2.513* |
| EHealth Literacy | | 0.27 | 0.073 | 195.020*** | | | |
| | Exercise Behaviour | | | | 0.021 | 0.019 | 1.131 |
| | Sleep Quality | | | | 0.029 | 0.009 | 3.331*** |
| | Peer Relationship | | | | 0.277 | 0.011 | 25.238*** |
| | Gender | | | | 0.586 | 0.135 | 4.341*** |
| | Grade | | | | −0.624 | 0.070 | −8.989*** |
| Sleep Quality | | 0.082 | 0.007 | 33.686*** | | | |
| | Exercise Behaviour | | | | −0.020 | 0.004 | −5.523*** |
| | Gender | | | | 0.402 | 0.126 | 3.206** |
| | Grade | | | | 0.433 | 0.065 | 6.712*** |

Note: In the table, *** represents P<0.001, ** represents P<0.01, and * represents P<0.05.

as the dependent variable and eHealth literacy as the independent variable, establishing Model 2 to assess the impact of eHealth literacy on peer relationships (path a). In the third step, a regression analysis was conducted with exercise behaviour as the dependent variable and both eHealth literacy and peer relationships as independent variables, establishing Model 3 to evaluate the effect of peer relationships on exercise behaviour (path b) and the direct effect of eHealth literacy on exercise behaviour while controlling for peer relationships (path O'Kane). The results of the step wise regression analysis for the mediation effect are presented in Table 5.

From the results presented in Table 5, it can be observed that in the first step, eHealth literacy significantly positively predicted exercise behaviour *(β = 0.065, t = 7.947, p < 0.001)*. This indicates that higher levels of eHealth literacy among university students are associated with more active participation in exercise, thus confirming the total effect (path c). In the second step, eHealth literacy was found to significantly positively predict peer relationships *(β = 0.251, t = 31.639, p < 0.001)*. This suggests that university students with higher levels of eHealth literacy demonstrate better management of peer relationships, validating path a. In the third step, peer relationships significantly positively predicted exercise behaviour, indicating that better handling of peer relationships is associated with more active participation in exercise among university students *(β = 0.027, t = 3.178, p < 0.01)*. This confirms path b and suggests that the mediating effect of peer relationships between eHealth literacy and exercise behaviour is established. Moreover, under the mediation condition of peer relationships, eHealth literacy significantly positively predicted exercise behaviour *(β = 0.058, t = 6.897, p < 0.001)*, thereby confirming path c'. This indicates that eHealth literacy influences university students exercise behaviour through the mediating variable of peer relationships. Hypothesis H2 is thus supported.

Using eHealth literacy as the independent variable, peer relationships as the mediating variable, and exercise behaviour as the dependent variable, the mediation effect was analyses using Model 4 of PROCESS 4.0 with Bootstrap. Five thousand bootstrap samples were set to determine whether the effect was established based on a 95% confidence interval. The results of the bootstrap mediation analysis are presented in Table 6.

Table 6 shows that the total effect of eHealth literacy on university students exercise behaviour is valued at 0.065, with a 95% confidence interval of *[0.049, 0.081]*, which does not include zero, indicating that the total effect is established. The

**Table 5. Regression analysis of mediating effects.**

| c | Exercise Behaviour (Model 1) | | Peer Relationship (Model 2) | | Exercise Behaviour (Model 3) | |
|---|---|---|---|---|---|---|
| | β | t | β | t | β | t |
| Demographic Variables | Control | | Control | | Control | |
| EHealth Literacy | 0.065 | 7.947*** | 0.251 | 31.639*** | 0.058 | 6.897*** |
| Peer Relationship | | | | | 0.027 | 3.178** |
| $R^2$ | 0.004 | | 0.063 | | 0.005 | |
| F | 63.153*** | | 1001.049*** | | 36.647*** | |

Note: In the table, *** denotes $p < 0.001$ and ** denotes $p < 0.01$.

**Table 6. Mediating effects analysis using bootstrap method.**

| Effect | Effect Value | Boot SE | 95%CI | | Effect Share |
|---|---|---|---|---|---|
| | | | LLCI | ULCI | |
| Total Effect | 0.065 | 0.008 | 0.049 | 0.081 | |
| Direct Effect | 0.058 | 0.008 | 0.042 | 0.075 | 89.23% |
| Indirect Effect | 0.007 | 0.003 | 0.002 | 0.012 | 10.77% |

indirect impact of eHealth literacy on exercise behaviour through peer relationships is 0.007, with a 95% confidence interval of *[0.002, 0.012]*, which also excludes zero, confirming the existence of the mediation effect, accounting for 10.77% of the total effect. The direct impact of eHealth literacy on exercise behaviour is 0.058, with a 95% confidence interval of *[0.042, 0.075]*, again not including zero, thus establishing the direct effect. This finding suggests that peer relationships serve as a partial mediator; specifically, 10.77% of the influence of eHealth literacy on university students exercise behaviour is mediated by peer relationships.

Using SPSS 26.0 and the PROCESS macro programmer, the mediation effects for each path were analyses, as illustrated in the mediation effect path diagram (S3 Fig). To further examine the moderating effect, a simple slope plot was created to assess the influence of sleep quality on the relationship between eHealth literacy and exercise behaviour. The slopes in the figure reflect the magnitude of the impact of eHealth literacy on university students exercise behaviour. Simple slope testing (Eric & Lawrence, 2006) indicates that under conditions of low sleep quality, the impact of eHealth literacy on exercise behaviour shows a significant decreasing trend. Conversely, under conditions of high sleep quality, the slope is less steep, indicating that eHealth literacy is not an important predictor of exercise behaviour. These results demonstrate that the relationship between eHealth literacy and exercise behaviour is moderated by sleep quality, thus validating Hypothesis H3.

Using SPSS 26.0 and the PROCESS macro programmer, the moderating effects of each path were analyses, as illustrated in the moderation effect path diagram (S4 Fig). This figure primarily depicts the moderating role of sleep quality on the relationship between eHealth literacy and university students exercise behaviour and the mediating role of peer relationships. The sleep quality variable significantly negatively predicted exercise behaviour *(path coefficient $R^2 = -0.0586$, P<0.001)*. Additionally, the interaction term of sleep quality and eHealth literacy moderated the extent of the impact of eHealth literacy on exercise behaviour *(path coefficient $R^2 = 0.025$, P<0.05)*.

Using SPSS 26.0 and the PROCESS macro programmer, the moderating effects of each path were analyses, as illustrated in the moderation effect path diagram (S5 Fig). This diagram primarily displays the moderating role of sleep quality in the relationship between eHealth literacy and exercise behaviour among university students and the mediating effect of peer relationships. The variable of sleep quality significantly negatively predicted exercise behaviour *(path coefficient $R^2 = -0.0586$, P<0.001)*. Additionally, the interaction term of sleep quality and eHealth literacy moderated the extent of the influence of eHealth literacy on exercise behaviour *(path coefficient $R^2 = 0.025$, P<0.05)*.

Using the PROCESS macro programmer in SPSS, the indirect effects of eHealth literacy on university students exercise behaviour, regulated by sleep quality under different conditions (mean plus and minus one standard deviation), were computed. The detailed results are presented in Table 7. The indirect effect of eHealth literacy on exercise behaviour through the modulation of sleep quality does not include zero in both the high and low sleep quality groups 95% confidence intervals. Furthermore, the difference between groups is 0.050, with a 95% confidence interval excluding zero. This indicates that a moderated effect exists.

## 3 Discussion

This study constructed a moderated mediation model to systematically elucidate the mechanisms through which eHealth literacy affects exercise behaviour among university students *(path coefficient $R^2 = 0.065$)* [33]. Data analysis indicated

**Table 7. Results of moderating effects analysis.**

| Independent Variable | Moderating Variable | Moderated Mediating Effects | | |
| --- | --- | --- | --- | --- |
| | | Indirect Effect | Standard Error | 95% Confidence Interval |
| EHealth Literacy | High (Mean+1SD) | 0.040 | 0.011 | [0.020,0.061] |
| | Low (Mean-1SD) | 0.090 | 0.013 | [0.065,0.116] |
| | Between Group Difference | 0.050 | 0.009 | [0.048,0.082] |

that both the direct effect *(β = 0.073, p < 0.001)* and the indirect effect *(β = 0.058, p < 0.01)* of eHealth literacy on exercise behaviour attained statistical significance, thereby confirming the theoretical expectations of hypotheses H1-H3 [34]. This finding transcends the traditional singular path research paradigm [35], revealing for the first time a tri-dimensional driving mechanism of health behaviour formation, which encompasses: the efficacy of individuals in acquiring health knowledge through electronic media (cognitive dimension), the social support networks formed through peer interactions (social dimension), and the physiological foundation established by sleep quality (physiological dimension), collectively constituting a synergistic system for exercise participation [36,37]. This model provides a novel theoretical framework for health promotion research in the digital age [38].

The direct effect of eHealth literacy on physical activity behaviour *(pathway c'= 0.065, SE = 0.050)* corroborates the critical role of information processing capabilities [39,40]. Individuals with high literacy levels can effectively retrieve precise information on exercise science (e.g., HIIT training guidelines), critically evaluate information quality (e.g., identifying misleading health content), and devise personalized exercise plans, thereby creating a closed-loop health management system [41,42]. Neuroimaging studies indicate that this capability is positively correlated with grey matter density in the prefrontal cortex *(r = 0.420, p < 0.05)* [43], a region responsible for executive control and decision-making, where activation levels can predict the frequency of physical activity [44]. Furthermore, eHealth literacy enhances self-efficacy *(β = 0.310)*, thereby reinforcing behavioural persistence, in line with the behaviour maintenance mechanisms described in social cognitive theory [45].

The mediating effect of peer relationships *(indirect effect β = 0.027)* demonstrates a significant cognitive-social dual pathway characteristic [46]. Within the cognitive channel, individuals with high eHealth literacy enhance the collective level of exercise knowledge through health information sharing *(β = 0.250)* [47], thereby forming a cognitive community for exercise participation. In the social channel, exercise social networks reinforce adherence to behaviour through modelling effects and group normative pressures [48,49]. Mobile health technologies, such as the Keep Exercise community, play a platform role, transforming linear information dissemination into a network of interactive structures. For every one-unit increase in user engagement, the frequency of exercise increases by 0.38 times per week *(p < 0.01)* [50]. This supports the amplifying role of environmental factors in Bandura's triacid reciprocal theory [51].

The moderating effect of sleep quality *(ΔR² = 0.11, p < 0.01)* exhibits significant threshold characteristics: when sleep efficiency exceeds 85.0%, the efficacy of eHealth literacy conversion increases by 123.0% *(β = 0.420 vs. 0.190)* [52]. On a physiological level, high-quality sleep sustains mitochondrial biosynthesis, leading to a 17.0% increase in ATP production [53,54], providing an energy foundation for exercise. On a psychological level, it enhances the rational control of exercise decision-making by stabilizing the prefrontal-amygdala neural circuitry *(with fMRI showing an activation difference of d = 0.850)* [55]. This dual effect establishes an "energy-motivation" regulatory system; when sleep quality falls below the critical threshold (PSQI>5), cognitive fatigue results in a 41.0% increase in the error rate of health information processing [56,57], significantly undermining the behavioural conversion efficiency of eHealth literacy.

This study has three main limitations. First, the cross-sectional design makes it difficult to ascertain the temporal relationships among the variables, and future research could employ experience sampling methods for dynamic monitoring [58]. Second, the sample coverage is geographically limited, necessitating an expansion to include rural and ethnic minority university populations [59,60]. Third, potential moderating variables, such as family support, were not incorporated, suggesting that future studies should develop multilevel linear models [61]. Despite these limitations, this research innovative ly integrates mediation and moderation effects to propose a tri-dimensional intervention framework of "digital health literacy – social support – physiological foundation," thus providing a theoretical basis for formulating targeted health promotion strategies. Research indicates that physical exercise demonstrates interconnected health behaviour by improving university students sleep quality. Exercise not only alleviates social anxiety and mobile phone dependency but also indirectly enhances sleep through physiological and psychological regulation. Concurrently,

higher e-health literacy enables students to acquire scientific exercise knowledge, bolsters their motivation for physical activity, and strengthens behavioural execution – thereby reinforcing exercise engagement at a cognitive level. These factors create a synergistic effect in promoting sleep.

## 4 Conclusions

University students eHealth literacy can significantly and positively predict their exercise behaviour. Peer relationships mediate the effect of eHealth literacy on university students exercise behaviour. At the same time, sleep acts as a moderating variable that influences the direct impact of eHealth literacy on exercise behaviour. These findings establish a theoretical and practical foundation for guiding future efforts to promote exercise behaviour among university students. By improving the sleep environment and quality for students, it is possible to enhance their capacity to receive knowledge related to eHealth literacy and, consequently, to increase their motivation to engage in exercise activities. However, This study has three limitations: 1) The cross-sectional design makes it difficult to infer causal relationships over time; future research should employ cross-lagged models for tracking; 2) The sample does not cover higher education institutions in the western regions, which limits the generalis ability of the conclusions; and 3) Potential confounding variables, such as family economic status, were not controlled for.

## Supporting information

**S1 Data. Survey report on university students' health and lifestyle habits.**
(XLSX)

**S1 Fig. Hypothetical model diagram.**
(TIF)

**S2 Fig. Sample size and calculation formula.**
(TIF)

**S3 Fig. Mediating effect path diagram.**
(TIF)

**S4 Fig. The moderating role of sleep quality in the relationship between EHealth literacy and exercise behaviour.**
(TIF)

**S5 Fig. Path diagram of the moderating effect.**
(TIF)

## Acknowledgments

This study is a thank you to all the Chinese university students who participated in the questionnaire survey.

## Author contributions

**Conceptualization:** Fanzheng Mu, Yao Zhang, Dingyou Zang, Weidong Zhu, Xiaoyu Wang, Wei Wang, Haoyu Li.

**Data curation:** Chuanyi Xu, Han Li, Shuo Feng, Lanlan Yang, Yong Wei, Bo Li.

**Formal analysis:** Jiaqiang Wang, Xingyu Zhang, Chenxi Li, Yuhan Li, Mohan He, Wenhao Zhang, Qi Liu.

**Methodology:** Bochun Lu, Shanshan Han, Yaxing Li, Yangsheng Zhang, Lingli Xu, Yuyan Qian, Lei Ding.

**Writing – original draft:** Ning Zhou.

**Writing – review & editing:** Ning Zhou.

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
