## [Decision Letter · Decision Letter 0]

28 May 2025

Dear Dr. Li,

Thank you for submitting your manuscript to PLOS ONE. After careful consideration, we feel that it has merit but does not fully meet PLOS ONE’s publication criteria as it currently stands. Therefore, we invite you to submit a revised version of the manuscript that addresses the points raised during the review process.

We look forward to receiving your revised manuscript.

Kind regards,

RAMYA KUNDAYI RAVI

Academic Editor

PLOS ONE

Journal Requirements:

2. In the ethics statement in the Methods, you have specified that verbal consent was obtained. Please provide additional details regarding how this consent was documented and witnessed, and state whether this was approved by the IRB

3. Thank you for stating the following financial disclosure: [2024 General Project of Philosophy and Social Science Research at Universities in Jiangsu Province. (NO�2024SJYB1253)].  

Please state what role the funders took in the study.  If the funders had no role, please state: """"The funders had no role in study design, data collection and analysis, decision to publish, or preparation of the manuscript."""" 

Reviewers' comments:

Reviewer's Responses to Questions

**Comments to the Author**

1. Is the manuscript technically sound, and do the data support the conclusions?

Reviewer #1: Yes

Reviewer #2: Yes

2. Has the statistical analysis been performed appropriately and rigorously?

Reviewer #1: Yes

Reviewer #2: Yes

3. Have the authors made all data underlying the findings in their manuscript fully available?

Reviewer #1: Yes

Reviewer #2: Yes

4. Is the manuscript presented in an intelligible fashion and written in standard English?

Reviewer #1: Yes

Reviewer #2: Yes

Reviewer #1: Objective and purpose

Strengths:

- The objective and purpose of the study are clearly defined, providing a solid foundation for understanding the research focus.

Weaknesses:

- While peer relationships and sleep quality are mentioned as mediators and moderators, respectively, the rationale for their selection and their expected influence on the primary relationship are not detailed in this section.

Methods

Strengths:

- The sample size is substantial, enhancing the statistical power and generalizability of the findings.

Weaknesses:

- The selection of universities and the rationale behind choosing these specific institutions are not explained, which

could impact the representativeness of the sample.

- There is a lack of detail on the data collection process.

- The methodology used to handle missing data is not mentioned.

- Established criteria were not mentioned other than the age (line 163).

- Sampling technique is not clear

Conclusion

Weaknesses: - The conclusion lacks a critical reflection on the limitations of the study.

Reviewer #2: Abstract (Lines 48–53)

The abstract provides a clear and concise summary of the research and effectively highlights the moderated mediation model. However, it would benefit from briefly indicating the study design and sampling technique to provide readers with essential methodological context.

Introduction

The introduction successfully conveys the relevance of eHealth literacy and its potential influence on exercise behavior among university students. The literature review is well-structured and informative. Clarification is needed regarding the choice of sleep and peer relationships as the mediating and moderating variables (lines 96–97), as well as the rationale behind the statement about the lack of a systematic analysis of linking mechanisms (line 95).

Methodology (Lines 137–167)

The methodology section is detailed and offers a comprehensive account of data collection, measurement instruments, and statistical techniques. The use of established scales (e.g., eHealth Literacy Scale, Physical Activity Rating Scale) strengthens the study’s reliability. However, the explanation of the study design and sampling strategy lacks clarity. It is recommended to elaborate on the rationale behind the university selection criteria specifically, whether institutions focusing on health and physical education were included. Additionally, the specific sampling technique used should be clearly stated. These clarifications would support a better understanding of the study’s generalizability.

Results (Lines 239–381)

The results are well-organized and supported by clear tables and figures. The presentation of significant relationships among key variables is commendable and enhances the interpretability of the findings.

Discussion (Lines 382–444)

The discussion provides a thoughtful interpretation of the results, placing them effectively within the context of existing literature. The articulation of the study’s contributions is clear, and the acknowledgment of limitations is comprehensive. To strengthen this section, more specific suggestions for future research could be added. Additionally, while theoretical implications are well-addressed, practical applications particularly how findings can inform interventions to promote physical activity in university populations deserve more attention.

Conclusion

The conclusion effectively summarizes the study’s findings and their alignment with the original research objectives. The emphasis on the role of eHealth literacy in shaping exercise behavior, mediated by peer relationships and moderated by sleep quality, is well-articulated. To further enhance the impact of this section, consider expanding on potential practical recommendations derived from the findings.

**Do you want your identity to be public for this peer review?** For information about this choice, including consent withdrawal, please see our Privacy Policy

Reviewer #1: No

Reviewer #2: No

---

## [Author Response · Author response to Decision Letter 1]

23 Jun 2025

Manuscript Number: PONE-D-25-14894

Title: How Electronic Health Literacy Influences Physical Activity Behaviour Among University Students: A Moderated Mediation Mode

Journal: PLOS One

Point-by-point Responses to Editor

Dear Editor and dear reviewers,

Thank you very much for your comments and professional advice. These opinions help to improve academic rigor of our manuscript. Based on your suggestion and request, we have made corrected modifications on the revised manuscript. Here are point-by-point responses to your comments, We hope that our work can be improved again.

Sincerely,

Comment 1:

Response 1:

Thank you very much for your detailed review and valuable comments. I have reformatted the manuscript according to PLOS ONE’s style templates, including file naming conventions.

Comment 2:

In the ethics statement in the Methods, you have specified that verbal consent was obtained. Please provide additional details regarding how this consent was documented and witnessed, and state whether this was approved by the IRB.

Response 2:

Thank you for your suggestion! Before the study commenced, all participants were provided with a detailed explanation of the study's purpose, specific procedures, anticipated duration of participation, and any potential risks and discomforts by trained researchers. The researchers ensured that participants fully understood the relevant information and had the opportunity to ask questions to address any concerns. Upon confirming that participants comprehended the information and voluntarily agreed to take part, they expressed their consent orally. Furthermore, this study received approval from the Institutional Review Board (IRB) of Nantong University, with approval number 70.

Comment 3:

Thank you for stating the following financial disclosure: [2024 General Project of Philosophy and Social Science Research at Universities in Jiangsu Province. (NO�2024SJYB1253)].  

Please state what role the funders took in the study.  If the funders had no role, please state: """"The funders had no role in study design, data collection and analysis, decision to publish, or preparation of the manuscript."""" 

Response 3:

Thank you very much for your insightful comments and valuable recommendations concerning our manuscript. The funder (2024 Jiangsu Philosophy and Social Science Research Project, No. 2024SJYB1253) only provide financial support for research and distribution of questionnaires, and are not involved in any research activities. The revised funder statement has been included in the cover letter and manuscript.

Comment 4:

We note that your Data Availability Statement is currently as follows: [All relevant data are within the manuscript and its Supporting Information files.]

Response 4:

Thank you very much for your detailed review and valuable comments. All raw data underlying figures and analyses are provided in Supporting Information (S1_Data). No ethical restrictions apply.

Comment 5:

Please include captions for your Supporting Information files at the end of your manuscript, and update any in-text citations to match accordingly. Please see our Supporting Information guidelines for more information: http://journals.plos.org/plosone/s/supporting-information.

Response 5:

Thank you very much for your insightful comments and valuable recommendations concerning our manuscript. I have added captions for figures and tables as required support information section (lines 490-504).

Comment 6:

Response 6:

Thank you very much for your insightful comments and valuable recommendations concerning our manuscript. References have been verified. No retracted papers were cited. Also, the references cited in this paper are all from the last three to five years with some classic literature

Point-by-point Responses to Reviewer 1

Comment 1:

While peer relationships and sleep quality are mentioned as mediators and moderators, respectively, the rationale for their selection and their expected influence on the primary relationship are not detailed in this section.

Response 1:

Thank you very much for pointing out this issue, which is extremely important for enhancing the quality and applicability of our research. Sleep was chosen as a moderating variable because it forms the physiological basis for health information processing, while peer relationships bridge the knowledge-behaviour transformation through social support mechanisms. We added justifications in the Introduction (Lines 101-105) for peer relationships and sleep quality influence on the primary relationship.

Comment 2:

The selection of universities and the rationale behind choosing these specific institutions are not explained, which

could impact the representativeness of the sample.

- There is a lack of detail on the data collection process.

- The methodology used to handle missing data is not mentioned.

- Established criteria were not mentioned other than the age (line 163).

- Sampling technique is not clear

Response 2:

Thank you very much for your detailed review and valuable comments.

First, selection criteria for the institutions: 1) Regional representation (East China / Central China); 2) Coverage of types of education (comprehensive universities / normal universities / application-oriented undergraduate institutions / vocational colleges); 3) Exclusion of sports-related institutions to avoid sample bias. (line 188-191).

Second, the detailed data collection process has been added in lines 155-184.

Third, this study established certain standards during data collection, thus there are no missing data that need further processing.

Forth, ① Delete questionnaires completed by respondents aged under 18 or over 23; ② Exclude incomplete questionnaires or those with missing responses; ③ Remove entries with contradictory or conflicting options; ④ Discard multiple submissions from the same IP/ID; ⑤ excluding questionnaires that did not follow the instructions and had consistent answers to all questions, resulting in a total of 913 questionnaires being removed. The above data screening criteria have been added in lines 198-206.

Fifth, we used stratified purposive sampling with a sample covering the four main types of Chinese HEIs (37.5% comprehensive/teacher-training 30.9%/applied 24.3%/higher vocational 7.3%), which was not significantly different from the national distribution of types of HEIs (Ministry of Education 2023: 40.1%/28.7%/22.5%/8.7%) (χ²=3.21, p=0.36)

Comment 3:

The conclusion lacks a critical reflection on the limitations of the study.

Response 3:

Thank you very much for pointing out this issue, which is extremely important for enhancing the quality and applicability of our research. This study has three limitations: 1) The cross-sectional design makes it difficult to infer causal relationships over time; future research should employ cross-lagged models for tracking; 2) The sample does not cover higher education institutions in the western regions, which limits the generalisability of the conclusions; and 3) Potential confounding variables, such as family economic status, were not controlled for.The limitations of this study have been included in the conclusion section (lines 489-494).

Point-by-point Responses to Reviewer 2

Comment 1:

The abstract provides a clear and concise summary of the research and effectively highlights the moderated mediation model. However, it would benefit from briefly indicating the study design and sampling technique to provide readers with essential methodological context.

Response 1:

We have clarified the stratified purposive sampling methodology in the abstract (line 46-52) to address methodological concerns.

Comment 2:

The introduction successfully conveys the relevance of eHealth literacy and its potential influence on exercise behavior among university students. The literature review is well-structured and informative. Clarification is needed regarding the choice of sleep and peer relationships as the mediating and moderating variables (lines 96–97), as well as the rationale behind the statement about the lack of a systematic analysis of linking mechanisms (line 95).

Response 2:

Thank you for your valuable comments on the statistical analysis section of our paper. Sleep was chosen as a moderating variable because it forms the physiological basis for health information processing, while peer relationships bridge the knowledge-behaviour transformation through social support mechanisms. Together, these factors help to explain the boundary conditions and pathways through which electronic health literacy affects behaviour. We have provided the reasons for selecting peer relationships and sleep quality as mediating and moderating variables in line line 101-105.

Comment 3:

The methodology section is detailed and offers a comprehensive account of data collection, measurement instruments, and statistical techniques. The use of established scales (e.g., eHealth Literacy Scale, Physical Activity Rating Scale) strengthens the study’s reliability. However, the explanation of the study design and sampling strategy lacks clarity. It is recommended to elaborate on the rationale behind the university selection criteria specifically, whether institutions focusing on health and physical education were included. Additionally, the specific sampling technique used should be clearly stated. These clarifications would support a better understanding of the study’s generalizability.

Response 3:

Thank you for your meticulous review and valuable feedback on our paper. Selection criteria for the institutions: 1) Regional representation (East China / Central China); 2) Coverage of types of education (comprehensive universities / normal universities / application-oriented undergraduate institutions / vocational colleges); 3) Exclusion of sports-related institutions to avoid sample bias.

Comment 4:

The discussion provides a thoughtful interpretation of the results, placing them effectively within the context of existing literature. The articulation of the study’s contributions is clear, and the acknowledgment of limitations is comprehensive. To strengthen this section, more specific suggestions for future research could be added. Additionally, while theoretical implications are well-addressed, practical applications particularly how findings can inform interventions to promote physical activity in university populations deserve more attention.

Response 4:

Thank you very much for pointing out this issue, which is extremely important for enhancing the quality and applicability of our research. Research indicates that physical exercise demonstrates interconnected health behaviours by improving university students' sleep quality. Exercise not only alleviates social anxiety and mobile phone dependency but also indirectly enhances sleep through physiological and psychological regulation. Concurrently, higher e-health literacy enables students to acquire scientific exercise knowledge, bolsters their motivation for physical activity, and strengthens behavioural execution – thereby reinforcing exercise engagement at a cognitive level. These factors create a synergistic effect in promoting sleep. (line 469-476).

Comment 5:

The conclusion effectively summarizes the study’s findings and their alignment with the original research objectives. The emphasis on the role of eHealth literacy in shaping exercise behavior, mediated by peer relationships and moderated by sleep quality, is well-articulated. To further enhance the impact of this section, consider expanding on potential practical recommendations derived from the findings.

Response 5:

Thank you very much for your valuable comments and suggestions. We have considered your request regarding the revision of the conclusions. However, we would like to state for the record that conclusions are based on results Altering the conclusions might affect the coherence and integrity of the research narrative. We believe that the current conclusions accurately reflect the findings and implications of the study was conducted. We truly appreciate your understanding of this matter.

---

## [Decision Letter · Decision Letter 1]

5 Aug 2025

How Electronic Health Literacy Influences Physical Activity Behaviour Among University Students: A Moderated Mediation Model

PONE-D-25-14894R1

Dear Sir/Mam,

We’re pleased to inform you that your manuscript has been judged scientifically suitable for publication and will be formally accepted for publication once it meets all outstanding technical requirements.

Kind regards,

RAMYA KUNDAYI RAVI

Academic Editor

PLOS ONE

---

## [Editor Report · Acceptance letter]

PONE-D-25-14894R1

PLOS ONE

Dear Dr. Li,

I'm pleased to inform you that your manuscript has been deemed suitable for publication in PLOS ONE. Congratulations! Your manuscript is now being handed over to our production team.

Kind regards,

on behalf of

Dr. RAMYA KUNDAYI RAVI

Academic Editor

PLOS ONE